# Boosting the efficiency of organic persistent room-temperature phosphorescence by intramolecular triplet-triplet energy transfer

Weijun Zhao[1], Tsz Shing Cheung[1], Nan Jiang[2], Wenbin Huang[2], Jacky W.Y. Lam[1,3], Xuepeng Zhang[1,3], Zikai He [2] & Ben Zhong Tang [1,3,4]

Persistent luminescence is a fascinating phenomenon with exceptional applications. However, the development of organic materials capable of persistent luminescence, such as organic persistent room-temperature phosphorescence, lags behind for their normally low efficiency. Moreover, enhancing the phosphorescence efficiency of organic luminophores often results in short lifetime, which sets an irreconcilable obstacle. Here we report a strategy to boost the efficiency of phosphorescence by intramolecular triplet-triplet energy transfer. Incorpotation of (bromo)dibenzofuran or (bromo)dibenzothiophene to carbazole has boosted the intersystem crossing and provided an intramolecular triplet-state bridge to offer a near quantitative exothermic triplet–triplet energy transfer to repopulate the lowest triplet-state of carbazole. All these factors work together to contribute the efficient phosphorescence. The generation and transfer of triplet excitons within a single molecule is revealed by low-temperature spectra, energy level and lifetime investigations. The strategy developed here will enable the development of efficient phosphorescent materials for potential high-tech applications.

[1] Department of Chemistry, The Hong Kong Branch of Chinese National Engineering Research Center for Tissue Restoration and Reconstruction, Institute of Molecular Functional Materials and The Hong Kong University of Science and Technology and the Institute for Advanced Study, Clear Water Bay, Kowloon, Hong Kong 999077, China. [2] School of Science, Harbin Institute of Technology, Shenzhen, HIT Campus of University Town, 518055 Shenzhen, China. [3] HKUST Shenzhen Research Institute, No. 9 Yuexing 1st RD, 518055 South Area, Hi-tech Park, Nanshan, Shenzhen, China. [4] Center for Aggregation-Induced Emission, SCUT-HKUST Joint Research Institutes, State Key Laboratory of Luminescent Materials and Devices, South China University of Technology, 510640 Guangzhou, China. These authors contributed equally: Weijun Zhao, Tsz Shing Cheung. Correspondence and requests for materials should be addressed to Z.H. (email: hezikai@hit.edu.cn) or to B.Z.T. (email: tangbenz@ust.hk)

Persistent luminescence is a fascinating phenomenon that luminophores continuously emit after the cease of excitation with lifetime of longer than 0.1 second[1,2]. Recent decades have witnessed the development of persistent luminescent materials for their exceptional applications in newly emerged technologies such as chemical sensor[3,4], anti-counterfeiting[5], optical recording[6], lighting and display[7], bio-imaging[8,9]. Compared to well-developed inorganic counterparts, organic materials showing persistent luminescence, such as organic persistent room-temperature phosphorescence (OPRTP), have also obtained significant breakthroughs in terms of lifetime-tuning benefiting from their structural versatility[10–14]. However, most OPRTP exhibit extremely low luminescence efficiency (Supplementary Figure 1 and 2), even some reported examples exhibit negligible spectral tails[15,16].

Enhancing the efficiency of OPRTP meets great challenge in modulating the long-lived triplet excitons which often endow complicated and multiple competitive decay channels[17–19]. The triplet excitons of organic molecules generally show inefficient and dim luminescence because of weak spin-orbit coupling and their high sensitivity to temperature, moisture, and molecular oxygen. On one hand, rigidification of the surrounding environment of phosphors and prevention of oxygen quenching are mostly utilized to facilitate the expected OPRTP property through the effective suppression of the radiationless decays ($k_{nr}$)[20–22] and quenching processes ($k_q$)[23–28]. On the other hand, tremendous efforts on manipulating intersystem crossing (ISC) have put forth several rational strategies to achieve an efficient spin-orbit coupling including heavy atom effect[29,30], El-Sayed's rule[31,32], energy gap principle[33–35]. However, the slow radiative decay ($k_P$) of OPRTP demandingly compete with the fast $k_{nr}$ and $k_q$ at ambient conditions. The heavy atom effect, El-Sayed's rule and energy gap principle also simultaneously contribute to high $k_P$ and $k_{nr}$ and efficient reverse ISC process to result in short lifetime. Hence, a conflict exists between lifetime prolongation and efficiency enhancement in OPRTP[32]. After our elucidation, we tried to solve the conflict by smart hybridizing excited states with (n,π*) and (π,π*) configurations to achieve balanced performance[32]. However, OPRTP materials still suffer from low phosphorescence quantum yield and this obstacle seems impossible to be solved by these strategies. Thus, design principle to boost the efficiency of OPRTP materials are highly demanded but challenging[36,37].

Triplet-triplet energy transfer (TTET) is a well-known photophysical process to exchange both the spin and energy between molecules or intramolecular fragments[38–42]. The efficient TTET process occurs when the two exchanging parts locate at a short distance (especially within one molecule with a distance of <10 Å) and release free energy (exothermic) via Dexter-type channel[39,43]. Fulfilling these requirements, we could effectively transfer the triplet excitons of the easily formed but short-lived donors to those of acceptors that are poorly formed but long-lived by molecular engineering. Such an approach could resolve the conflict to provide a strategy to obtain efficient OPRTP materials.

Carbazole derivatives are promising luminescent materials for their high quantum yields ($\Phi$) and stability. Carbazole represents one of the earliest OPRTP materials in the crystalline state with a lifetime ($\tau$) of up to several seconds. However, the phosphorescence quantum yield ($\Phi_P$) contributes little to the overall $\Phi$[44]. Thanks to their lowest triplet excited state with pure $^3(\pi,\pi^*)$ configuration, phosphorescent carbazole derivatives have been found to inherit persistent lifetimes[45,46]. However, their $\Phi_P$ are still far-reaching from the efficiency requirement.

In this article, we integrated (bromo)dibenzofuran or (bromo) dibenzothiophene groups with carbazole to create twisted phosphors. Efficient ISC process is rationally designed and promoted with the aid of heavy atom, small energy gap and spin-vibronic

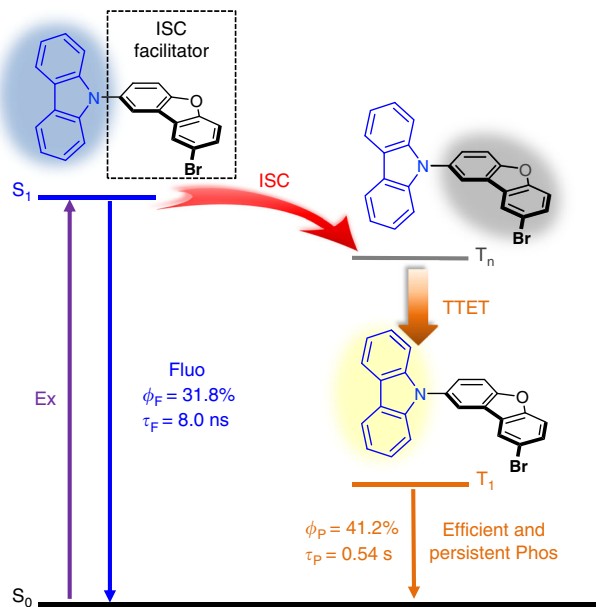

**Fig. 1** Strategy to achieve efficient OPRTP by intramolecular TTET process. Representation of the excited-state decay pathways in bromodibenzofuran-substituted carbazole, lifetime, and efficiency in the crystalline state

coupling mechanism[47]. A near quantitative TTET process occurs to carry the triplet excitons from the high-lying (bromo)dibenzofuran or (bromo)dibenzothiophene groups back to carbazole to emit persistent RTP. Highly efficient and persistent phosphors are generated with the best $\Phi_P$ of up to 41% (73% for total $\Phi$) and lifetime of 0.54 s under ambient conditions. The rational design strategy developed here should be also applicable for exploring OPRTP materials with high-tech applications.

## Results

**Material design**. The design strategy is presented schematically in Fig. 1. This representation illustrates the prominent photophysical processes and the relative energy levels. For proof-of-concept, carbazole was selected as the photo-absorption chromophore and also the luminophore where the lowest singlet ($S_1$) and triplet ($T_1$) states are located with bromodibenzofuran as the ISC facilitator. This integrated molecule endows a twisted conformation and partially charge-transfer excited states. The suitable ordering of energy level, small singlet-triplet energy gap, and heavy bromine atom leads efficient ISC and TTET processes. Dual emission with a fluorescence quantum yield ($\Phi_F$) of 31.8% and persistent phosphorescence quantum yield ($\Phi_P$) of 41.2% are observed.

To further validate our proposal, four integrated molecules, namely CZ-DBF, CZ-DBFBr, CZ-DBT, and CZ-DBTBr were facilely prepared through C–N coupling reaction between carbazole (CZ) and dibenzofuran (DBF), bromodibenzofuran (DBFBr), dibenzothiophene (DBT), and bromodibenzothiophene (DBTBr), respectively (Fig. 2a and Supplementary Figure 3-19). As purity is crucial for photophysical property, we purified them successively by column chromatography and recrystallization for three times. Elemental analysis and high performance liquid chromatography are applied to check their purity (Supplementary Figure 20). We check the change of photoluminescent (PL) spectra and transient decay profiles of the molecules during the recrystallization process and found that they were almost the same (Supplementary Figure 21). This suggests that the phosphors have high purity for photophysical property investigation.

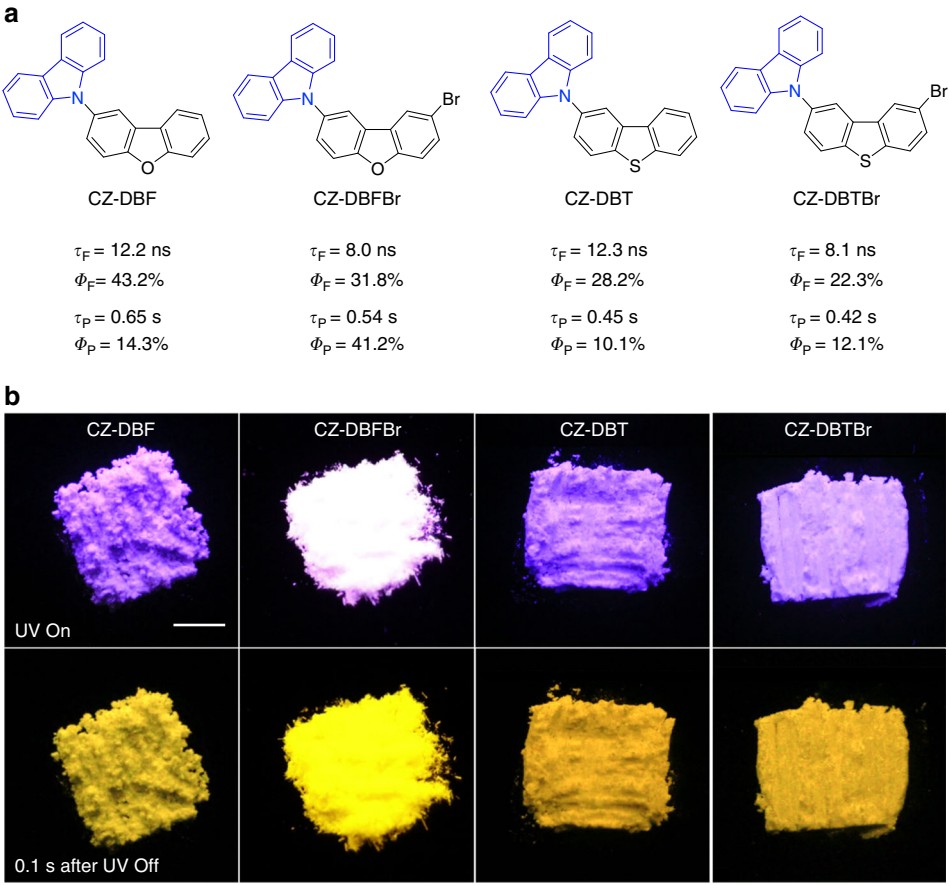

**Fig. 2** Chemical structure and luminescent performance of OPRTP compounds. **a** Molecular structures of OPRTP materials with dual fluorescence and phosphorescence lifetimes and quantum yields. **b** Photographs of crystalline powders taken before and after removal of UV excitation source of 365 nm at ambient conditions. Scale bar = 0.5 cm

As expected, all the designed molecules show efficient and persistent emission with performance summarized in Fig. 2a. CZ-DBFBr for example, exhibits bright and persistent white emission (Fig. 2b). However, its model fragments CZ and DBFBr only show prompt blue fluorescence (Supplementary Figure 22). After removing the UV irradiation, the emission of CZ-DBFBr lasts for seconds viewed by naked eyes. Similarly, CZ-DBF, CZ-DBT, and CZ-DBTBr also exhibit intense persistent emission (Fig. 2b), which is different from their prompt fluorescent fragments (Supplementary Figure 23 and 24).

**Photophysical property**. As shown in Fig. 3a-d, all the crystalline powders of the integrated molecules exhibit dual fluorescence and phosphorescence at ambient conditions[48]. The similarly shaped steady-state (prompt) PL spectra (blue zones) cover two independent emission bands of 380−520 nm and 520−710 nm with varied ratios. The red zones describe the time-resolved (delayed for 100 ms) PL spectra, where the prompt emission bands at 380−520 nm disappear completely for their short lifetimes. The perfect overlapping between the red zones and the emission bands at 520−710 nm reveals its persistent phosphorescence characteristic.

To verify the emission nature, the transient decay profiles at 430 and 550 nm were recorded (Fig. 3e, f). The emission at 430 nm shows lifetimes in the nanosecond scale in accordance with fluorescence. On the other hand, the emission at 550 nm shows lifetimes in the second scale and is undoubtedly the persistent phosphorescence. In detail, CZ-DBF and CZ-DBT exhibit dual emission with $\Phi_F$ of 43.2% ($\tau_F = 12.2$ ns) and 28.2%

($\tau_F = 12.3$ ns), along with $\Phi_P$ of 14.3% ($\tau_P = 0.65$ s) and 10.1% ($\tau_P = 0.45$ s), respectively. With an additional bromine atom, CZ-DBFBr and CZ-DBTBr show weaker fluorescence with smaller $\Phi_F$ of 31.8% ($\tau_F = 8.0$ ns) and 22.3% ($\tau_F = 8.1$ ns) but stronger phosphorescence with higher $\Phi_P$ of 41.2% ($\tau_P = 0.54$ s) and 12.1% ($\tau_P = 0.42$ s), respectively. Thus, the presence of bromine atoms enhances the $\Phi_P$ but shorten the lifetime due to the well-known heavy atom effect.

Interestingly, CZ-DBFBr exhibits single molecular white-light emission by suitably mixing the balanced fluorescence and phosphorescence bands with the Commission Internationale de l'Éclairage 1931 chromaticity coordinates of (0.34, 0.30) (Supplementary Figure 25). It is worth to note that the two emission bands endow the vibronic features of the CZ unit and cover the same wavelength range, suggesting that the four compounds have quite similar $S_1$ and $T_1$ states from CZ unit. The slight difference should come from the aggregation effect of the crystalline states.

**Energy level ordering**. To decipher the working mechanism, the energy levels of $S_1$ and $T_1$ states of the model fragments and the integrated molecules were carefully estimated from their low-temperature fluorescence and phosphorescence spectra. The well vibration-resolved profiles allow accurate energy level abstractions based on the 0−0 peaks for deaerated solution samples in 2-methyl-tetrahydrofuran at 77 K (Supplementary Figure 26) and solid crystalline samples at 4 K (Supplementary Figure 27). The persistent lifetimes are also measured (Supplementary Figure 28 and Supplementary Table 1-3). As summarized in Fig. 4, the

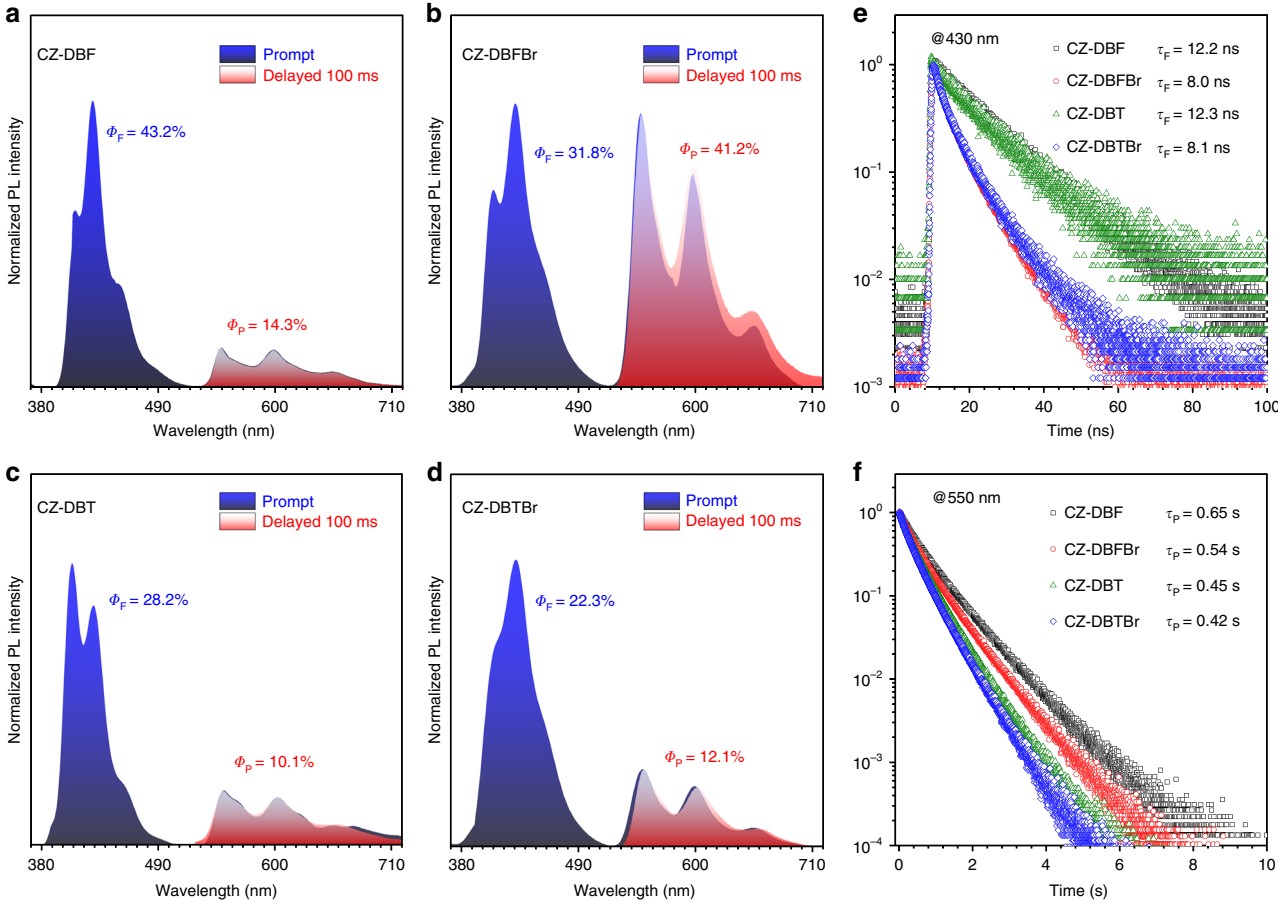

**Fig. 3** Photophysical properties of CZ-DBF, CZ-DBFBr, CZ-DBT, and CZ-DBTBr. **a–d**, The prompt (blue zone) and delayed (red zone, 100 ms) PL spectra of the crystalline powders of CZ-DBF (**a**), CZ-DBFBr (**b**), CZ-DBT (**c**), and CZ-DBTBr (**d**) with $\Phi_F$ and $\Phi_P$ inset. Note that the red zones are perfectly overlapped with the longer ranges of the blue zones. **e, f**, Transient PL decay curves of CZ-DBF, CZ-DBFBr, CZ-DBT, and CZ-DBTBr measured at 430 nm (**e**) and 550 nm (**f**) with $\tau$ inset. The excitation was 365 nm

integrated molecules basically have energy levels and lifetimes inherited from the corresponding low-lying model fragments.

In rigid-glass solutions at 77 K, all the model fragments exhibit fluorescence and phosphorescence emission as the nonradiative decay pathways are effectively suppressed (Supplementary Figure 26). DBFBr and DBTBr emit dominant phosphorescence with short lifetime because of the heavy atom effect, and they are thus good ISC facilitators. The integrated molecules emit accordingly. CZ-DBF perfectly inherits the $S_1$ and $T_1$ states from CZ, while CZ-DBFBr, CZ-DBT, and CZ-DBTBr have lower $S_1$ and $T_1$ energy levels than those of CZ-DBF. The sophisticated vibration-resolved phosphorescence spectra in solution states suggests that the $T_1$ states are locally-excited ($^3$LE) originated from CZ in CZ-DBF, DBFBr in CZ-DBFBr, DBT in CZ-DBT, and DBTBr in CZ-DBTBr, respectively. They endow the lowest triplet energy levels[49]. The charge-transfer ($^1$CT) characteristic is revealed by the red-shifted shoulder peak in UV-vis absorption spectra (Supplementary Figure 29 and 30) and solvent-dependent featureless emission in the PL spectra[50] (Supplementary Figure 31) which explains the lower $S_1$ energy levels.

In the crystalline state at 4 K, the model fragments emit red-shifted fluorescence and weaker phosphorescence due to the host environmental change (Supplementary Figure 27). For example, the bright fluorescence of CZ shifts from 345 to 405 nm with a $\Phi_F$ of 78.2% and the dim phosphorescence shifts from 406 to 549 nm from the solution state to crystalline state, with significantly decreased energy levels of $S_1$ and $T_1$. As a result, the integrated

molecules all show quite similar PL (wavelength) spectra (Fig. 3a-d, Supplementary Table 1 and 3), with both $S_1$ and $T_1$ inherited from the CZ units with LE nature. Even at room temperature, the integrated molecules show fluorescence and phosphorescence mainly from the CZ unit, while crystalline powders of DBF and DBFBr already lost their detectable phosphorescence (Supplementary Figure 27).

As well known, the room-temperature phosphorescence of CZ solid is extremely low, but the efficient phosphorescence of the integrated molecules all stems from the CZ units in the crystalline state. What happens to the CZ unit after its integration with other fragments into one molecule? The solid evidences that other fragments are strong ISC facilitators and CZ always endows with lowest $S_1$ and $T_1$ energy levels suggest a possible two-step photophysical process of integrated molecules excitons. This includes the efficient ISC from CZ to the attached fragments followed by the complete TTET to CZ giving both emissive $S_1$ and $T_1$ states.

**Intersystem crossing and crystal packing.** Intersystem crossing plays a vital role in obtaining efficient OPRTP. For the phosphors investigated here, it is rationally designed and promoted as follows: heavy atom effect in bromo-substituted derivatives with a corresponding higher $\Phi_P$; small singlet-triplet energy gaps of $0.17-0.23$ eV in the crystalline state where DBF or DBT fragments provide $T_n$ triplet bridges; boosted ISC between $S_1$ and $T_n$ by $^3$LE state mediated spin-vibronic coupling mechanism. Recently, the

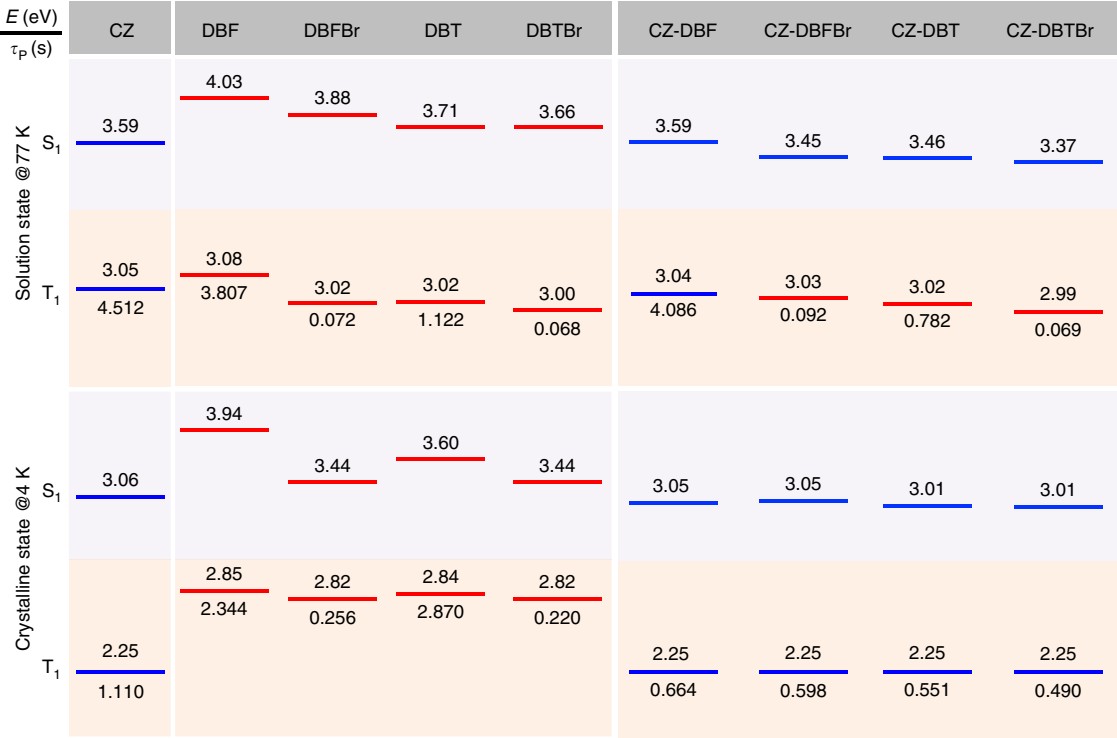

**Fig. 4** Energy levels and lifetimes of the model fragments and integrated molecules. The energy levels were calculated from the 0–0 peaks or the onsets of the emission bands

second-order spin-vibronic mechanism successfully reveals the efficient reverse ISC process in the thermally-activated delayed fluorescence (TADF) molecules with donor–acceptor (D–A) structure[47,51]. An intermediate triplet-state, either the $^3$LE or $^3$CT state depending on the rigidity and polarity of the host, involves the nonadiabatic coupling and thermal equilibrium between the lowest triplet states $^3$LE-$^3$CT. The coupling provides either the donor or acceptor $^3$LE to mediate the efficient reverse ISC between almost degenerated $^1$CT and $^3$CT states. The energy levels of $^3$LE and CT then critically controls the efficiency of this mechanism.

Here, the observation of weak to no fluorescence and strong phosphorescence in solution at 77 K reveals that boosted ISC occurs in CZ-DBF, CZ-DBFBr, CZ-DBT, and CZ-DBTBr with the aid of favored spin-vibronic coupling mechanism. In solution at 77 K, $S_1$ states with variable $^1$CT characteristic and $T_1$ states with low-lying $^3$LE characteristic are revealed based on the sophisticated vibration-resolved phosphorescence spectra (Supplementary Figure 29-31). Nearly pure phosphorescence emission indicates the high to unity ISC efficiency based on Ermolev's rule[52]. The direct spin-orbit coupling of low-lying $^3$LE and $^1$CT and the vibronic-promoted second-order coupling of almost degenerate $^3$CT and $^1$CT are considered to facilitate the efficient ISC process.

In the solid state, the $S_1$ states change their characteristic from $^1$CT to $^1$LE on the CZ units and the $T_n$ states are now localized on other $^3$LE model fragments. The ISC process becomes less efficient as the total PL spectra change from nearly pure phosphorescence emission (Supplementary Figure 26) in the solution to main fluorescence emission (Supplementary Figure 32 and 33) in the solid state when nonradiative decays are significantly suppressed at these low temperatures. An endothermic vibronic-coupling $^3$CT state above the two $^3$LE states is required to mediate the before-mentioned spin-vibronic coupling, which seems quite difficult with the high rigidity of the crystalline state[53,54]. Meanwhile, the direct ISC transition between the two LE state is also unfavorable because of poor orbital-overlapping

and low vibrational Franck-Condon factor, resulting in the weak yet moderate ISC process. Nevertheless, the reduced rate constants of $k_{ISC}$ are still high up to over $5.15 \times 10^7$ s$^{-1}$, which is comparable to the high ISC rate of TADF molecules (Supplementary Table 3 and Supplementary Note 1).

Single crystals of CZ-DBF, CZ-DBFBr, and CZ-DBT qualified for X-ray crystallography are grown from slow evaporation of their dichloromethane/hexane solutions. Attempts to grow the single crystals of CZ-DBTBr for X-ray crystallography were tried but failed as the obtained crystalline needles too thin for analysis. The crystal structure of CZ is also provided for comparison. The details of their crystal structures are given in Supplementary Table 4-6. As shown in Fig. 5, the CZ units pack in Herringbone modes in all crystals without π-to-π coplanar interactions. Such a packing mode should contribute to the aggregation effect that significantly decreases the energy levels of $S_1$ and $T_1$ excitons of CZ. The dihedral angles between the model fragments are depicted. The large dihedral angles of 49° to 72° suggest a twist structure of the molecules with poor conjugation and a high chance to form CT excited states[55,56].

**Triplet-triplet energy transfer.** Boosted ISC allows migration of abundant triplet excitons from high-lying $T_n$ state to CZ-based $T_1$ state through quantitative TTET. The process is investigated by time-resolved excitation/emission spectra and temperature-dependent steady-state/transient PL spectra. Taking CZ-DBFBr as an example, the time-resolved excitation spectra of CZ, DBFBr, and CZ-DBFBr are measured by monitoring the persistent emission bands in the crystalline state at 4 K (Fig. 6a). The obvious red-shifted spectra inherited from CZ units allow selective excitation of the CZ-DBFBr fragments. Irradiated at the longer wavelength of 400 nm, only the CZ units are excited. The time-resolved persistent (delayed for 100 ms) emission spectra of CZ, DBFBr, and CZ-DBFBr are then recorded (Fig. 6b and Supplementary Figure 32-34). Different from the results measured at room

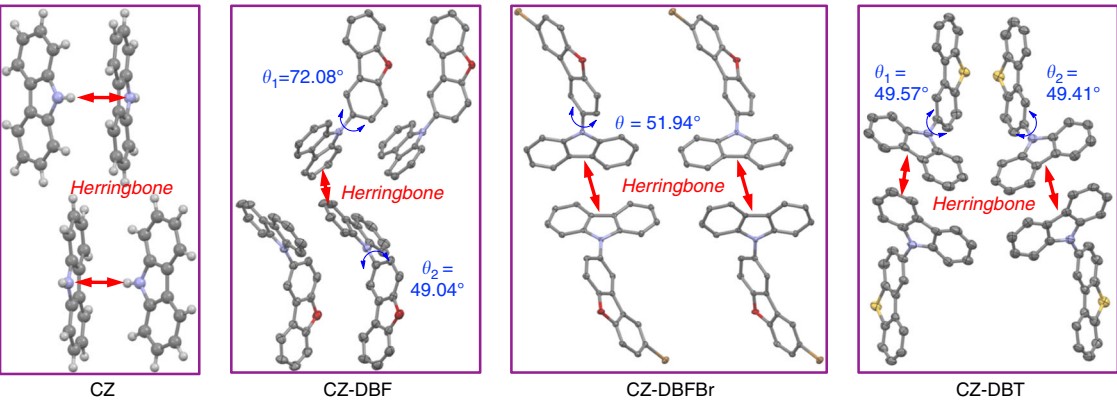

**Fig. 5** Crystal structures of CZ, CZ-DBF, CZ-DBFBr, and CZ-DBT. Carbon, hydrogen, oxygen, nitrogen, sulfur, and bromine atoms are shown as gray, white, red, blue, yellow, and orange balls (for CZ) or ellipsoids (for CZ-DBF, CZ-DBFBr, and CZ-DBT) at the 50% probability level

**Fig. 6** Time-resolved excitation/emission and temperature-dependent PL spectra. **a** Time-resolved excitation spectra of CZ, DBFBr, and CZ-DBFBr in the crystalline state at 4 K. **b** Time-resolved emission spectra of CZ, DBFBr, and CZ-DBFBr in solid state at 4 K. **c** Temperature-dependent steady-state PL spectra of CZ-DBFBr in solid state from 4 to 300 K. **d** Temperature-dependent intensity of emission bands at 500 and 655 nm. The excitation of CZ and CZ-DBFBr was 400 nm; while the excitation of DBFBr was 330 nm

temperature as shown in Fig. 2b, the observed persistent phosphorescence of CZ-DBFBr is clearly sum of CZ and DBFBr emission in vibrational resolution. This indicates the existence of triplet excitons of CZ and DBFBr fragments. In detail, the emission band at 440−530 nm solely stems from DBFBr triplet excitons. As the triplet energy level of DBFBr is higher than that of CZ at this extremely low temperature of 4 K, the triplet excitons of DBFBr should only originate through exothermic ISC process from the selectively excited CZ $S_1$ excitons.

The temperature-dependent steady-state and transient PL spectra reveal the relationship between the CZ and DBFBr triplet excitons (Supplementary Figure 35). With increasing the temperature from 4 to 200 K, the intensity of the emission band at 500 nm band is almost quenched, but the intensity of the emission band at 655 nm band increases by almost two-fold (Fig. 6c, d). These data imply that the triplet excitons of DBFBr quickly transfer to low-lying intramolecular CZ unit through Dexter-type TTET channel at elevated temperatures, which is similarly to internal conversion through thermal vibrational relaxation. When the temperature was further increased from 200 to 300 K, the emission band at 500 nm disappeared and the emission intensity at 655 nm was also partially decreased due to the near-quantitatively occurred TTET process and enhanced nonradiative decay pathways. The overall PL spectra then exhibit only CZ emission by eliminating other fragments' contribution. The temperature-dependent transient PL decay profiles are plotted in Supplementary Figure 35 and the corresponding lifetimes are shown in Supplementary Figure 36. The $\tau_P$ at 500 nm experiences a continuous decrease from 4 to 200 K while $\tau_P$ at 655 nm keeps almost unchanged. The quite similar trend was also found for the lifetime of CZ, but different for DBFBr fragment (Supplementary Figure 37). It suggests the increased nonradiative decay of DBFBr triplet excitons through the accelerated TTET pathway in the CZ-DBFBr.

## Discussion

We previously elucidated the conflict of the lifetime prolongation and efficiency enhancement in OPRTP materials. By smart hybridizing excited states with (n,π*) and (π,π*) configurations, OPRTP materials with balanced performances have been achieved[32]. Now people begin to realize this obstacle in design of OPRTP and pay great attention to this issue[36,37]. On one hand, if $k_{nr}$ and $k_q$ are suppressed to an ignorable level, it is possible to achieve high efficiency. However, such possibility is extremely challenging under ambient conditions. On the other hand, if the influence on $k_{ISC}$ and $k_P$ is reversed, we may simultaneously achieve high $k_{ISC}$ for enhancing the efficiency and maintain the $k_P$ to a low level for prolonging the lifetime. TTET plays such a role here. The triplet excitons of the donors are sufficiently achieved but short-lived and are effectively transferred to the acceptors, whose triplet excitons are poorly achieved but long-lived[57]. The energy level determines the transfer direction. Moreover, when the triplet excitons are populated through ISC, they come to the emissive $T_1$ states in a near quantitative yield through TTET. Thus, TTET successfully separate the locations of the initially-populated $T_n$ center and the finally-emissive $T_1$ center. In comparison, blended powders by melting the mixture of CZ and DBFBr exhibit negligible RTP spectral tails, suggesting that ISC or TTET is much less efficient than the emerged single molecules (Supplementary Figure 38).

In summary, we put forth a design strategy to boost the efficiency of OPRTP by intramolecular TTET. Four (bromo)dibenzofuran or (bromo)dibenzothiophene substituted carbazoles are designed and show efficient and persistent luminescence with the best $\Phi_P$ of up to 41% and lifetime of 0.54 s under ambient conditions. Efficient ISC process is promoted with the aid of heavy atom, small energy gap and spin-vibronic coupling mechanism. The nearly quantitative TTET occurs from the high-lying (bromo)dibenzofuran or (bromo) dibenzothiophene group to the carbazole unit. The rational design strategy developed here would be applicable for exploring OPRTP materials with high-tech applications.

## Methods

**Crystal growth.** All the crystalline samples were obtained from slowly evaporative crystallization using hexane/dichloromethane mixture (2:1, v/v). To further check the purity of the solid samples, all the crystalline samples were dissolved in 100% acetonitrile and got sample solutions (50 μM), then run the HPLC.

## Data Availablity

The authors declare that the all data supporting the findings of this study are available within this article and Supplementary Information files, and also are available from the authors upon reasonable request. CCDC 1865076, 1865078, 1865082 contain the crystallographic data for CZ-DBF, CZ-DBFBr, and CZ-DBT in this paper. These data can be obtained free of charge from The Cambridge Crystallographic Data Centre via www. ccdc.cam.ac.uk/getstructures.

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

## Acknowledgements

The authors acknowledge the financial support by the National Science Foundation of China (21788102 and 51703042), the University Grants Committee of Hong Kong (AoE/P-03/08), the Science and Technology Plan of Shenzhen (JCYJ20160229205601482 and JCYJ20170811155015918), the Research Grants Council of Hong Kong (16308016, 16305015, C6009-17G, C2014-15G and A-HKUST605/16), and the Innovation and Technology Commission (ITC-CNERC14SC01 and ITS/254/17).

## Author contributions

Z.H. and B.Z.T. designed and supervised the research and wrote the paper. W.Z. and T.S. C. synthesized all materials and grew the crystals. W.Z. performed all the photophysical measurements. Thus, T.S.C. and W.Z. contributed equally to this work. N.J., W.H. and X. Z. assisted the photophysical property measurement. J.W.Y.L revised and edited the manuscript. All authors discussed the results and commented on the manuscript.

## Additional information

**Competing interests:** The authors declare no competing interests.

