## [Peer Review File · Nature Communications]

Reviewers' comments:

Reviewer #1 (Remarks to the Author):

Comments:

In this work, the authors have put forth a novel strategy to design metal-free organic phosphors with efficient and persistent RTP feature facilitated by intermolecular TTET process. The performance is really impressive considering the obstacle of enhancing the efficiency and prolonging the lifetime for RTP. Detailed photophysical experiments were performed to disclose the intrinsic structure-property relationship including excited energy levels, luminescent spectra, phosphorescence lifetime, intersystem crossing and TTET, etc. The data are comprehensive and the conclusions are reliable. It puts forward a new train of thought on RTP research, which is general, inspirable and applicable. As a result, this is an elaborate piece of work and thus recommended to be published on Nature Communications after some minor revisions:

1. In Figure 1, the excited-state decay pathways are related to the RTP molecule at aggregate state which should be illustrated in the caption.
2. The authors integrated carbazole and triplet bridged fragments in a single molecular to facilitate the intersystem crossing and realize the efficient triplet energy transfer. To better evaluate this strategy, the author should investigate the performance of blended fragments.
3. In the experiments of temperature dependent phosphorescence lifetime (Figure 6). The caption of the inset figure in Figure 6d should be added. And, the authors should study the temperature dependent phosphorescence lifetime of each fragment (CZ and DBFBr) for a more reliable conclusion.
4. Why the authors demonstrate that TTET process is quantitatively occurred?
5. Several parameters of RTP and triplet energy transfer in this work may be compared with other related references (such as Nat. Commun., 2018, 9, 2798 ; Chem. Sci. 2016, 7, 4519).

Overall, in my view, this work could arouse broad interest in chemistry and fluorescent material (particularly for recent hot topic RTP) fields, and thus is suitable for publication in Nature Communications after some revision.

Reviewer #2 (Remarks to the Author):

The authors report on a new strategy to achieve persistent phosphorescence at room temperature with strong yield. I find the work very interesting and in general well done. However, there are several points that need to be clarified before the paper can be accepted for publication. Therefore, I recommend the manuscript to be rejected in this current version.

- 1) the idea of using D-A molecules to achieve strong RTP emission is not entirely new. It is true the authors express here in a more focused way than in previous works the rationale to use the energy alignment between the singlet state and a higher triplet state to promote ISC, followed by IC to a low lying triplet state from the phosphorescence occurs. However, this has been already reported in the literature, see for example the work of Dias et al. (DOI: 10.1039/c8tc02987c). This should be at least referenced.
- 2) The authors refer several times to the spin-vibronic mechanism to explain the intense phosphorescence. I encourage the authors to explain the physics of this mechanism and not just using jargon, what is meant by spin-vibronic coupling and how it works?
- 3) Is this spin-vibronic mechanism still operative at 77 K in 2Me-THF? If yes, then the ISC should be larger at RT than at 77 K, right? If not, why is the ISC close to 100% at 77 K as the authors

claim?

4) P9L9. Why is the ISC between the two LE states localized on the D and A units less efficient than between 1CT and 3LE?

5) P9L10. An endothermic spin-vibronic coupling seems incompatible with the high rigidity of the crystalline state to explain the "weakened yet moderate" ISC. What is the evidence to support the authors claim that the ISC is weaker in the crystal phase, when compared to solution.

6) P8L24. A value for the ISC rate of Cz is given. How was this value obtained?

7) P9L6. The authors explain that in solution near 100% ISC is obtained. How was the ISC efficiency measured?

8) P9L12. Once more a value for the ISC rate is given, but how was it measured? (in fact is incorrect, as explained below).

9) Finally, the values for the ISC rate and yield in Table 3 are incorrect. The equation from where k_{ISC} is determined is wrong. Φ_{IP} and Φ_{ISC} are not the same thing. Actually Φ_{IP} depends on Φ_{ISC} .

Minor points:

P2L4 "The triplet excitons of...because of their high sensitivity..." sensitivity to what?

P2L20 references on etc. Please allocate references to well identified facts.

Fig 10a in SI. The time scale appears to be incorrect.

Several mistakes on the solvent nomenclature on Fig 13 in SI.

Reviewer #3 (Remarks to the Author):

The authors report the high efficiency persistent phosphorescence. By employing intramolecular triplet transfer, both the efficiency (41.2%) and lifetime (0.54 s) are improved compared with their last report (Chem 1, 592-602, 2016) of 36% and 0.23 s.

However, there are some problems need to be clarified, which may improve the quality of this manuscript.

1. In the abstract, can the authors explain the "normally low efficiency"? Does it mean the "low efficiency" is normal and high efficiency is abnormal? And why the lifetime and efficiency are irreconcilable, as described by the author in the following sentence?

2. As illustrated in Figure 1, the two fragments share the same ground state but exhibit their own excited states, S1, Tn and T1. The authors propose that the phosphorescence is from the CZ unit solely but through ISC and TTET process. Therefore, the emission spectra show similar vibronic characters of CZ unit. Is the vibronic states from excited states or ground states? And how the emerged ground states affect this?

3. On Page 8, "they endow the lowest triplet energy levels". But the lowest triplet at 77K for CZ-DBFBr and CZ-DBT and CZ-DBTBr are not CZ component. How the authors explain this with the emission from CZ?

4. Is the high efficiency recorded from powder? The authors studied the energy states at 77K at diluted solution (assume single molecular state) and at 4K at crystalline state. How these two conditions relate to their powder form?

5. On page 9, "the large dihedral angle.....providing the chance to form the CT excited states". The authors should provide reference here for such claim.

6. The authors should refine their language. For example, 'Early we elucidate the conflict.....' "Now people realize the"; "redder" page 8.

Responses to the Comments and Suggestions of Reviewer 1

The reviewer commented that our work “*put forth a novel strategy to design metal-free organic phosphors with efficient and persistent RTP feature*” and evaluated that our work “*arouse broad interest in chemistry and luminescent material (particularly for recent hot topic RTP) fields, and thus is suitable for publication in Nature Communications after some revision*”. We thank the reviewer for his/her recognition of this work. He/she also pointed out several minor issues of the manuscript. We revised the manuscript accordingly. Below are our point-to-point responses to the reviewer’s comments and suggestions.

Comment 1: *In Figure 1, the excited-state decay pathways are related to the RTP molecule at aggregate state which should be illustrated in the caption.*

Our replay: Thanks for your careful revision. As recommended, we add a more accurate illustration to the caption of Figure 1 as follows: “**Representation of the excited-state decay pathways in bromodibenzofuran-substituted carbazole, lifetime and efficiency at the crystalline state.**”

Comment 2: *The authors integrated carbazole and triplet bridged fragments in a single molecular to facilitate the intersystem crossing and realize the efficient triplet energy transfer. To better evaluate this strategy, the author should investigate the performance of blended fragments.*

Our replay: Thanks for your thoughtful suggestions. We conducted the additional experiments which are shown in Supplementary Figure 20 accordingly. The blended fragments only exhibit negligible RTP spectral tails, suggesting that ISC or TTET is much less efficient in the blended systems than in the emerged single molecules.

We also include the description of blending experiments and results in our manuscript as follows: “**Blended powders by melting the mixture of CZ and DBFBr exhibit negligible RTP spectral tails, suggesting that ISC or TTET is much less efficient than the emerged single molecules (Supplementary Figure 20).**”

Supplementary Figure 20. The prompt (black line) and delayed (red line) PL spectra of the blended powders of CZ and DBF (a), CZ and DBFBr (b), CZ and DBT (c), CZ and DBTBr (d) at 300 K. The excitation wavelength was 365 nm.

Comment 3: *In the experiments of temperature dependent phosphorescence lifetime (Figure 6). The caption of the inset figure in Figure 6d should be added. And, the authors should study the temperature dependent phosphorescence lifetime of each fragment (CZ and DBFBr) for a more reliable conclusion.*

Our replay: Thanks for your careful revision. We add the caption of inset figure in Figure 6d as follows: “**d, Temperature-dependent intensity of 500 and 655 nm emission bands and with lifetime inset.**” We conducted the suggested experiments as shown below. When the temperature is enhanced from 4 to 300 K, lifetimes of CZ and DBFBr decreased slowly due to the increased nonradiative decay without TTET process.

We also include the description of blending experiments and results in our manuscript as follows: “**The quite similar trend was also found for the lifetime of CZ, but different for DBFBr fragment. (Supplementary Figure 19) It suggests the increased nonradiative decay channel of DBFBr triplet excitons through the accelerated TTET pathway in the integrated CZ-DBFBr molecule.**”

Supplementary Figure 19. Temperature-dependent phosphorescence lifetimes of CZ and BDFBr.

Comment 4: *Why the authors demonstrate that TTET process is quantitatively occurred?*

Our replay: Thanks for your careful revision. As show in Figure 6c and d, we demonstrate that “When the temperature was further increased from 200 to 300 K, the emission band at 500 nm disappeared and the emission intensity at 655 nm was also partially decreased due to the near-quantitatively occurred TTET process and enhanced nonradiative decay pathways.”. Therefore, TTET process should quantitatively occur for the emerged compounds at room temperature.

Comment 5: *Several parameters of RTP and triplet energy transfer in this work may be compared with other related references (such as Nat. Commun., 2018, 9, 2798; Chem. Sci. 2016, 7, 4519).*

Our replay: Thanks for your valuable suggestion. We tried our best to summarize the parameters of reported RTP compounds in Supplementary Figure 1. More necessary references have been added to our revised manuscript.

Responses to the Comments and Suggestions of Reviewer 2

The reviewer commented our work “*report a new strategy to achieve persistent phosphorescence at room temperature with strong yield*” and find “*the work very interesting and in general well done*”. We are gratefully for your appreciation of our work and thanks for your precious time to review our manuscript. We also thank your professional and valuable comments. Accordingly, we revise the manuscript point by point as follows.

Comment 1: *the idea of using D-A molecules to achieve strong RTP emission is not entirely new. It is true the authors express here in a more focused way than in previous works the rational to use the energy alignment between the singlet state and a higher triplet state to promote ISC, followed by IC to a low lying triplet state from the phosphorescence occurs. However, this has been already reported in the literature, see for example the work of Dias et al. (DOI: 10.1039/c8tc02987c). This should be at least referenced.*

Our Reply: Thanks for your professional revision. The idea of using D-A molecules to achieve strong RTP emission is not entirely new. Just pointed by the reviewer, here we are focusing in the rational design strategy of utilizing TTET process between a high triplet state and a low lying triplet state to promote the whole ISC process from S1 to T1, in which the D-A molecules are employed as the demonstrations. Therefore, we still maintain the novelty of the work. During our preparation of the manuscript, a related work of Dias et al. (DOI: 10.1039/c8tc02987c) as mentioned by the reviewer was published. **So, we cite the work in our revised manuscript accordingly.**

Comment 2: *The authors refer several times to the spin-vibronic mechanism to explain the intense phosphorescence. I encourage the authors to explain the physics of this mechanism and not just using jargon, what is meant by spin-vibronic coupling and how it works?*

Our Reply: Thanks for your professional revision. We give a more detailed explanation of the spin-vibronic mechanism in our revised manuscript accordingly as follows:

“Recently, the second order spin-vibronic mechanism successfully reveals the efficient reverse ISC process in the thermally-activated delayed fluorescence (TADF) molecules with donor-acceptor (D-A) structure^{47,51}. An intermediate triplet state, either the ³LE or ³CT state

depending on the rigidity and polarity of the host, involves the nonadiabatic coupling between the lowest triplet states ^3LE - ^3CT . The coupling provides either the donor or acceptor ^3LE to mediate the efficient reverse ISC between almost degenerated ^1CT and ^3CT states. The energy levels of ^3LE and CT then critically controls the efficiency of this mechanism.”.

Comment 3: *Is this spin-vibronic mechanism still operative at 77 K in 2Me-THF? If yes, then the ISC should be larger at RT than at 77 K, right? If not, why is the ISC close to 100% at 77 K as the authors claim?*

Our Reply: Thanks for your thoughtful considerations. Yes, this spin-vibronic mechanism still works at 77 K in 2Me-THF, similar to the Type I TADF in the *Nat. Commun.* work of Monkman (DOI: 10.1038/ncomms13680) where $E^3_{\text{LE}} < E_{\text{CT}}$. The direct spin-orbit coupling of low-lying ^3LE and ^1CT and the vibronic-promoted second-order coupling of almost degenerate ^3CT and ^1CT are considered to facilitate the efficient ISC process.

The ISC should be larger at room temperature than that at 77 K because the k_{isc} is increased due to enhanced spin-vibronic effect. However, the radiationless quenching and internal conversion pathways are also enhanced at room temperature, giving the highly competitive nonradiative rate constants. Therefore, only fluorescence emission is observed in the aerated solutions at RT. The degassed solutions exhibit noticeable but not dominated phosphorescence emission (See Figure below) as triplet excitons are highly sensitive to temperature, moisture and molecular oxygen than singlet excitons. Therefore, the overall result is that the efficiency of ISC at RT in solution is slightly decreased when compared to 77 K for the greatly enhanced fluorescent decay and radiationless process.

The fact that the ISC is close to 100% is based on the observation of weak to no fluorescence in total PL spectra at 77 K in solutions and other considerations as detailed in **Comment 7**.

Figure Steady-state emission spectra in toluene solution at RT, in air and in nitrogen for CZ-DBF (**a**), CZ-DBFBr (**b**), CZ-DBT (**c**) and CZ-DBTBr (**d**). The excitation wavelength was 330 nm.

Comment 4: *P9L9. Why is the ISC between the two LE states localized on the D and A units less efficient than between 1CT and 3LE ?*

Our Reply: There are several reasons that the ISC between the two LE states localized on the D and A units less efficient than between 1CT and 3LE , including the energy gap, spin-orbit coupling, etc. The direct ISC transition between the two LE state is unfavorable probably because of poor orbital-overlapping (dramatically different electron distribution on the D and A units) and low vibrational Frank-Condon factor according to the Fermi's golden rule.

Comment 5: P9L10. *An endothermic spin-vibronic coupling seems incompatible with the high rigidity of the crystalline state to explain the “weakened yet moderate” ISC. What is the evidence to support the authors claim that the ISC is weaker in the crystal phase, when compared to solution?*

Our Reply: Thanks for your professional revision. After careful analysis of the ISC process, we agree with the reviewer. An endothermic spin-vibronic coupling seems incompatible with the high rigidity of the crystalline state. Therefore, we remove this part to explain the “weakened yet moderate” ISC. We revised our manuscript as follows:

An endothermic vibronic-coupling ^3CT state above the two ^3LE states is required to mediate the before-mentioned spin-vibronic coupling, which seems quite difficult with the high rigidity of the crystalline state^{53,54}. Meanwhile, the direct ISC transition between the two LE state is also unfavorable because of poor orbital-overlapping and low vibrational Franck-Condon factor, resulting in the weak yet moderate ISC process. Nevertheless, the reduced rate constants of k_{ISC} are still high up to over $5.15 \times 10^7 \text{ s}^{-1}$, which is comparable to the highest ISC rate of TADF molecules (Supplementary Table 3).

The claim that *the ISC is weaker in the crystal phase when compared to solution* is based on the experimental fact that the total PL spectra change from nearly pure phosphorescence emission (Supplementary Figure 9) in the solution to main fluorescence emission (Supplementary Figure 16) in the solid state when nonradiative decays are significantly suppressed at these low temperatures.

Comment 6: P8L24. *A value for the ISC rate of Cz is given. How was this value obtained?*

Comment 8: P9L12. *Once more a value for the ISC rate is given, but how was it measured? (in fact is incorrect, as explained below)?*

Our Reply: Thanks for your careful revision. After further analysis the data, we also find some problems about the ISC rate of Cz. The calculated value for others ISC rates is also incorrect. To avoid the misleading, we revised these two parts as follows:

“As well known, the room temperature phosphorescence of CZ solid is extremely low”.

“The rate constants of k_{ISC} are enhanced up to over $0.82 \times 10^7 \text{ s}^{-1}$, which is comparable to the high ISC rate of TADF molecules (Supplementary Table 3).”.

The following are our detailed reasons.

First, the equations we referenced in the previous supporting information are incorrect. So we revised our demonstrations in the Supplementary Table 3 and manuscript. Theoretically, the lifetime (τ_F and τ_P) and photoluminescence quantum yield (Φ_F and Φ_P) of fluorescence and RTP as well as quantum yield of ISC (Φ_{ISC}) can be expressed as follows.

$$\tau_F = 1 / (k_F + k_{nr}^F + k_{ISC}) \quad (1)$$

$$\tau_P = 1 / (k_P + k_{nr}^P) \quad (2)$$

$$\Phi_F = k_F / (k_F + k_{nr}^F + k_{ISC}) = k_F \times \tau_F \quad (3)$$

$$\Phi_P = \Phi_{ISC} \times k_P / (k_P + k_{nr}^P) = \Phi_{ISC} \times k_P \times \tau_P \quad (4)$$

$$\Phi_{ISC} = k_{ISC} / (k_{ISC} + k_F + k_{nr}^F) = k_{ISC} \times \tau_F \quad (5)$$

In our previous references (*Nat. Mater.* **14**, 685-690 (2015) and *Adv. Mater.* **30**, 1803856 (2018)), the authors assumed that the non-radiative rate constants of the triplet state (k_{nr}^P) is much lower than radiative rate constants of the triplet state (k_P). Actually, the value of k_{nr}^P is comparable with the value of k_P . Therefore, we consider that referenced assumption is incorrect. We can easily obtain Equation (6) from Equation (2):

$$\Phi_{ISC} = \Phi_P (1 + k_{nr}^P / k_P) \quad (6)$$

From Equation (6) and (4), we know that the value of Φ_{ISC} should be higher than Φ_P but lower than $1 - \Phi_F$. Therefore, k_{ISC} can be estimated by Φ_{ISC} / τ_F , accordingly.

Second, we consider that k_{nr}^P / k_P should be lower than $1 / \Phi_P - 1$. As emerged compounds show similar lowest triplet characters of CZ at similar crystalline packing modes, we assume that they should have the similar level of k_{nr}^P / k_P . According to our experiment data of τ_P , we can get that k_{nr}^P / k_P is lower than 5.99, 1.43, 8.90 and 7.26 for CZ-DBF, CZ-DBFBr, CZ-DBT and CZ-DBTBr, respectively. Therefore k_{nr}^P / k_P of CZ should be around 10^1 . Furthermore, as detection limit of luminescence quantum yield is about 0.01%, Φ_P of CZ

should be lower than 0.01% based on the fact of undetectable phosphorescence emission. According to the above two assumptions, we can get that Φ_{ISC} of CZ should be lower than 0.1% and k_{ISC} of CZ should be lower than $1.22 \times 10^5 \text{ s}^{-1}$.

Last, to avoid the misleading and accurate the demonstration, we still remove the description of ISC rate constant of CZ in the revised manuscript.

Comment 7: P9L6. *The authors explain that in solution near 100% ISC is obtained. How was the ISC efficiency measured?*

Our Reply: Thanks for your careful revision.

According to fundamental photophysics, the high rigidity of glass solution samples by cooling to a very low temperature of 77 K and/or polymer matrix/crystalline samples can effectively eliminate the radiationless quenching and internal conversion pathways. So, k_{nr}^F is considered much lower than k_F at these conditions. This is perfectly true for aromatic hydrocarbons and other rigid systems, known as the Terenin's Rule and Ermolev's Rule. By the way, considering the fact that the phosphorescence is very strong while the fluorescence is found weak to no in solution state at 77 K (Supplementary Figure 9), we assume k_{ISC} is much higher than k_F at low temperature. Therefore, we demonstrated the yield of ISC is near 100% according to equation (7) for CZ-DBTBr:

$$\Phi_{ISC} = \frac{k_{ISC}}{k_{ISC} + k_{nr}^F + k_F} \quad (7)$$

As shown in revised Supplementary Figure 9a-c, the proportion of phosphorescence to the PL of CZ-DBF, CZ-DBFBr and CZ-DBT is 69.1%, 91.1% and 96.4%, respectively, which are much higher than CZ (42.8%). Therefore, we demonstrated that “In solution at 77 K, S_1 states with variable 1CT characteristic and T_1 states with low-lying 3LE characteristic are revealed based on the sophisticated vibration-resolved phosphorescence spectra (Supplementary Figure 12-14). Nearly pure phosphorescence emission indicates the high to unity ISC efficiency based on Ermolev's rule⁵².”

Supplementary Figure 9. The prompt (line) and delayed (color zone) PL spectra of the 2-methyl-tetrahydrofuran solutions of CZ, DBF and CZ-DBF (a), DBFBr and CZ-DBFBr (b), DBT and CZ-DBT (c), DBTBr and CZ-DBTBr (d) at 77 K with the proportion of phosphorescence inset. Excitation wavelength: 290 nm.

Comment 9: Finally, the values for the ISC rate and yield in Table 3 are incorrect. The equation from where k_{ISC} is determined is wrong. Φ_{iP} and Φ_{iISC} are not the same thing. Actually Φ_{iP} depends on Φ_{iISC} .

Our Reply: Thanks for your careful revision.

Actually, we cannot get the accurate value of ISC from our experiment data. Our explanation is similar to **Comment 6**, to avoid the misleading, we revised the Supplementary Table 3 as follow:

Supplementary Table 3. Photoluminescence wavelength (λ_F and λ_P), lifetime (τ_F and τ_P) and quantum efficiency (Φ_F and Φ_P), rate constant of ISC (k_{isc}) of crystalline CZ-DBF, CZ-DBFBr, CZ-DBT and CZ-DBTBr.

Compound	Temp (K)	λ_F (nm) ^a	λ_P (nm) ^a	τ_F (ns)	τ_P (ms)	Φ_F (%)	Φ_P (%)	Φ_{ISC} (%)	k_{ISC} ($10^7 s^{-1}$) ^b
CZ-DBF	4	407	548	12.6	664	43.2	14.3	56.8 > Φ_{ISC} > 14.3	4.51 > k_{ISC} > 1.17
	300	411	549	12.2	652				
CZ-DBFBr	4	407	548	8.1	598	31.8	41.2	68.2 > Φ_{ISC} > 41.2	8.52 > k_{ISC} > 5.15
	300	411	550	8.0	540				
CZ-DBT	4	412	550	14.5	551	28.2	10.1	71.8 > Φ_{ISC} > 10.1	5.84 > k_{ISC} > 0.82
	300	413	551	12.3	450				
CZ-DBTBr	4	412	550	8.2	490	22.3	12.1	77.7 > Φ_{ISC} > 12.1	9.59 > k_{ISC} > 1.49
	300	413	551	8.1	420				

^a 0-0 peak of the photoluminescence wavelength.

^b Theoretically, the lifetime (τ_F and τ_P) and photoluminescence quantum yield (Φ_F and Φ_P) of fluorescence and RTP as well as quantum yield of ISC (Φ_{ISC}) can be expressed as follows.

$$\tau_F = 1 / (k_F + k_{nr}^F + k_{ISC}) \quad (1)$$

$$\tau_P = 1 / (k_P + k_{nr}^P) \quad (2)$$

$$\Phi_F = k_F / (k_F + k_{nr}^F + k_{ISC}) = k_F \times \tau_F \quad (3)$$

$$\Phi_P = \Phi_{ISC} \times k_P / (k_P + k_{nr}^P) = \Phi_{ISC} \times k_P \times \tau_P \quad (4)$$

$$\Phi_{ISC} = k_{ISC} / (k_{ISC} + k_F + k_{nr}^F) = k_{ISC} \times \tau_F \quad (5)$$

We can easily obtain Equation (6) from Equation (2):

$$\Phi_{ISC} = \Phi_P (1 + k_{nr}^P / k_P) \quad (6)$$

From Equation (6), we know that the value of Φ_{ISC} should be higher than Φ_P but lower than $1 - \Phi_F$. Therefore, k_{ISC} can be estimated by Φ_{ISC} / τ_F accordingly.

Minor points:

P2L4 “The triplet excitons of....because of their high sensitivity...” sensitivity to what?

Our Reply: Thanks for your careful revision.

We follow the reviewer’s consideration and revised our manuscript as follows: “the weak spin-orbit coupling and their high sensitivity to temperature, moisture and molecular oxygen.”

P2L20 references on etc. Please allocate references to well identified facts.

Our Reply: Thanks for your suggestion. We have allocated references (Ref. 3-9) to well identified facts.

Fig 10a in SI. The time scale appears to be incorrect.

Our Reply: Thanks for your careful revision. We have changed the time scale “s” to “ns”.

Several mistakes on the solvent nomenclature on Fig 13 in SI.

Our Reply: Thanks for your careful revision. We have revised the solvent nomenclature on Supplementary Figure 13 and 14.

Responses to the Comments and Suggestions of Reviewer 3

The reviewer commented our work “*report the high efficiency persistent phosphorescence. By employing intramolecular triplet transfer, both the efficiency (41.2%) and lifetime (0.54 s) are improved*”. He/she also pointed out several small issues which need to be clarified to improve the quality of this manuscript. We thank the reviewer for his/her recognition of this work and the nice advices he/she made, and we revised the manuscript accordingly. Below are our point-to-point responses to the reviewer’s comments.

Comment 1: *In the abstract, can the authors explain the “normally low efficiency”? Does it mean the “low efficiency” is normal and high efficiency is abnormal? And why the lifetime and efficiency are irreconcilable, as described by the author in the following sentence?*

Our replay: Thanks for your careful revision.

On one hand, to confirm our demonstration we summarized the properties of reported persistent RTP compounds in Supplementary Figure 1. By screening their RTP performance, we conclude that “**most OP RTP exhibit extremely low luminescence efficiency (Supplementary Figure 1), even some reported examples exhibit negligible spectral tails.**” in the introduction part of the manuscript.

On the other hand, when using the same rigidification methodology to suppress k_{TS} to almost the same level, the increase in lifetime will simultaneously decrease the efficiency according to the equation: $\Phi_P = \Phi_{ISC}(1 - k_{TS} \tau_P)$. Therefore, a slow k_P is needed for achieving a persistent lifetime but cannot easily compete with k_{TS} and will thus have a low efficiency. A high k_P will lead to a short lifetime but high efficiency as a result of the better competition with k_{TS} . As a result, endowing a persistent RTP system with high efficiency becomes more difficult than for systems with short lifetimes. **We have also included such demonstration in our discussion part with proper references.**

Comment 2: *As illustrated in Figure 1, the two fragments share the same ground state but exhibit their own excited states, S1, Tn and T1. The authors propose that the phosphorescence is from the CZ unit solely but through ISC and TTET process. Therefore, the emission spectra show similar vibronic characters of CZ unit. Is the vibronic states from*

excited states or ground states? And how the emerged ground states affect this?

Our replay: Thanks for your professional revision.

The vibronic characters should stem from the ground states based on the Franck-Condon principle, that the emission comes from transitions of the $\nu = 0$ vibrational level of the lowest excited states to the different vibrational levels of the ground states. Considering the vertical transition of locally excited states of S_1 and T_1 on CZ units, we still observed quite similar vibronic characters of CZ units, which are slightly influenced by the aggregation effect of crystalline state.

As you mentioned, the emerged ground states of integrated molecules definitely affect the vibronic features of emission spectra, such as peak wavelength, intensity and shape etc. which are dominated by CZ units.

We add more description of this part as follows: “It is worth to note that the two emission bands endow the vibronic features of the CZ unit and cover the same wavelength range, suggesting that the four compounds have quite similar S_1 and T_1 states from CZ unit. The slight difference should come from the aggregation effect of the crystalline states.”

Comment 3: *On Page 8, “they endow the lowest triplet energy levels”. But the lowest triplet at 77 K for CZ-DBFBr and CZ-DBT and CZ-DBTBr are not CZ component. How the authors explain this with the emission from CZ?*

Our replay: Thanks for your careful revision.

In this part, we investigated the excited energy levels of samples both in solution state and in crystalline (solid) state. As you said, the lowest triplet energy levels of CZ-DBFBr and CZ-DBT and CZ-DBTBr at 77 K for are not CZ component **in solution state**. As shown in Figure 4 (solution state @77 K), supplementary Figure 9 and Table 2, we demonstrated that “The sophisticated vibration-resolved phosphorescence spectra suggests the T_1 states are locally-excited (3LE), which originate from CZ in CZ-DBF, DBFBr in CZ-DBFBr, DBT in CZ-DBT and DBTBr in CZ-DBTBr, respectively. They endow the lowest triplet energy levels.”

However, they endow the same lowest triplet energy levels **in solid state**. In the crystalline state at 4 K and room temperature, the integrated molecules all show quite similar PL

(wavelength) spectra (Figure 3a-d, Supplementary Table 1 and 3), with both S_1 and T_1 inherited from the CZ units with LE nature. We demonstrated that the CZ units pack in Herringbone modes in all crystals as shown in Figure 5. Such a packing mode should contribute to the aggregation effect that significantly decreases the energy levels of S_1 and T_1 excitons of CZ.

To avoid misleading, we change the sequence of this part in the revised manuscript.

Comment 4: *Is the high efficiency recorded from powder? The authors studied the energy states at 77 K at diluted solution (assume single molecular state) and at 4 K at crystalline state. How these two conditions relate to their powder form?*

Our replay: Thanks for your careful revision.

The powders shown in the photos of Figure 2b are crystalline samples with small size. We slightly ground the big single crystals into smaller size for measurement convenience. Therefore, we add detailed demonstration in the caption of Figure 2b as follows: “**Photographs of crystalline powders** taken before and after the removal of excitation source of UV light at ambient conditions”.

All the photophysical measurements of solid samples are recorded from their same crystalline powders at different temperatures.

Comment 5: *On page 9, “the large dihedral angle.....providing the chance to form the CT excited states”. The authors should provide reference here for such claim.*

Our replay: Thanks for your good suggestion. We have added two related references in the revised manuscript.

Chen, C. et al. Intramolecular Charge Transfer Controls Switching Between Room Temperature Phosphorescence and Thermally Activated Delayed Fluorescence. *Angew. Chem. Int. Ed.* 57, 16407-16411 (2018).

Etherington, M. K. et al. Regio- and conformational isomerization critical to design of efficient thermally-activated delayed fluorescence emitters. *Nat. Commun.* 8, 14987, (2017).

Comment 6: *The authors should refine their language. For example, ‘Early we elucidate the conflict.....’ “Now people realize the”; “redder” page 8.*

Our replay: Thanks for your good suggestion.

We have asked a colleague whose native language is English to review our manuscript. We have also carefully checked the English throughout the manuscript.

REVIEWERS' COMMENTS:

Reviewer #1 (Remarks to the Author):

In this revised manuscript, the authors have carefully answered all the questions point-to-point from the reviewers, and I am pleased to recommend to accept this work as it is.

Reviewer #2 (Remarks to the Author):

I am now satisfied with the revisions made by the authors, in particular with the corrections made to estimate the ISC rates. This is a good work and I, therefore, recommend publication.

F.B. Dias